# Incidence and Fate of Refractory Type II Endoleak after EVAR: A Retrospective Experience of Two High-Volume Italian Centers

**DOI:** 10.3390/jpm12030339

**Published:** 2022-02-24

**Authors:** Pasqualino Sirignano, Nicola Mangialardi, Martina Nespola, Francesco Aloisi, Matteo Orrico, Sonia Ronchey, Flavia Del Porto, Maurizio Taurino

**Affiliations:** 1Vascular and Endovascular Surgery Unit, Sant’Andrea Hospital, Department of Surgery Paride Stefanini, Sapienza University of Rome, 00189 Rome, Italy; 2Department of Vascular Surgery, Ospedale San Camillo-Forlanini, 00152 Rome, Italy; nikmangialardi@gmail.com (N.M.); dr.orrico.matteo@gmail.com (M.O.); sonia.ronchey@gmail.com (S.R.); 3Vascular and Endovascular Surgery Unit, Sant’Andrea Hospital, Department of Molecular and Clinical Medicine, Sapienza University of Rome, 00189 Rome, Italy; martina.nespola@gmail.com (M.N.); fra.aloisi@gmail.com (F.A.); maurizio.taurino@uniroma1.it (M.T.); 4Internal Medicine Unit, Sant’Andrea Hospital, Department of Molecular and Clinical Medicine, Sapienza University of Rome, 00189 Rome, Italy; flavia.delporto@uniroma1.it

**Keywords:** abdominal aortic aneurysms, AAA, endovascular repair, EVAR, type 2 endoleaks, refractory endoleaks, reintervention, conversion, semiconversion

## Abstract

Introduction: The aim of the present study is to report the outcome of patients presenting an isolated type II endoleak (TIIEL) requiring reintervention and to identify clinical and anatomical characteristics potentially implicated in refractory TIIEL occurrence and fate. Materials and Methods: A multicenter retrospective study on TIIEL requiring reintervention was conducted between January 2003 and December 2020. Demographic and clinical characteristics, procedural technical aspects, reinterventions, and outcomes were recorded. TIIEL determining sac expansion greater than 10 mm underwent a further endovascular procedure aiming to exclude aneurismal sac. Redo endovascular procedures were performed via endoleak nidus direct embolization and/or aortic side branches occlusion. TIIELs responsible for persisting aneurysmal sac perfusion 6 months after redo endovascular procedures were classified as “refractory” and submitted to open conversion. Results: A total of 102 TIIEL requiring reintervention were included in the final analysis. Eighty-eight (86.27%) patients were male, the mean age was 77.32 ± 8.08 years, and in 72.55% of cases the American Society of Anaesthesiologists (ASA) class was ≥3. The mean aortic diameter was 64.7 ± 14.02 mm, half of treated patients had a patent inferior mesenteric artery (IMA), and 44.11% ≥ 3 couples of patent lumbar arteries (LA). In 49 cases (48.03%) standard endovascular aneurysm repair (EVAR) procedure was completed without adjunctive maneuvers. All enrolled patients were initially submitted to a further endovascular procedure once TIIEL requiring reintervention was diagnosed; 57 patients underwent LAs or IMA embolization (55.87%), 42 transarterial aneurismal sac embolization (41.17%), and three (2.96%) laparoscopic ostial ligations of the inferior mesenteric artery. During a mean follow-up of 15.22 ± 7.57 months (7–48), a redo endovascular approach was able to ensure complete sac exclusion in 52 cases, while 50 patients presented a still evident refractory TIIEL and therefore a surgical conversion or semiconversion was conducted. At the univariate analysis refractory TIIEL patients were significantly different from those who did not develop the complication in terms of preoperative clinical, morphological characteristics, and initial EVAR procedures: coronary artery disease occurrence (*p* = 0.005, OR: 3.18, CI95%: 1.3–7.2); preoperative abdominal aortic aneurysm (AAA) sac diameter (*p* = 0.0055); IMA patency (*p* = 0.016, OR: 2.64, CI95%: 1.18–5.90); three or more patent LAs; isolated standard EVAR without adjunctive procedures (*p* > 0.0001; OR: 9.48, CI95%: 3.84–23.4). Conclusions: Our experience seems to demonstrate that it is reasonable to try to preoperatively identify those patients who will develop a refractory TIIEL after EVAR and those with a TIIEL requiring reintervention for whom a simple endovascular redo will not be enough, needing surgical conversion.

## 1. Introduction

Endovascular aneurysms repair (EVAR) has been widely accepted as the procedure of choice for patients with infrarenal abdominal aortic aneurysm (AAA) not only in suitable but also in complex anatomies [1].

Nevertheless, EVAR still presents the concern of frequent reinterventions for long-term complications, primarily endoleaks [2,3,4,5,6,7]. Among them, type 2 endoleak (TIIEL) is the most reported cause of reintervention after standard EVAR [8]. TIIEL, defined by the persistent perfusion of the aneurysmal sac after EVAR by side branches, such as lumbar arteries (LAs) and/or the inferior mesenteric artery (IMA), presents a variable incidence that can be as high as 40% [9].

About 60% of the immediate postoperative TIIELs heal spontaneously, without sac enlargement, and no need for further treatments [10]. However, the fate of persistent TIIELs is variable, as the aneurysm sac size may decrease, remain stable, or increase. Notwithstanding, TIIEL is generally considered a benign disease, at least if it is not associated with significant sac expansion [10]. Although sac expansion of more than 5 mm has been reported to be associated with AAA rupture after EVAR [11], a recent review showed that almost 30% of patients with TIIEL evolving towards rupture do not have any documented sac expansion.

In contrast to type I and type III endoleaks, for which need for intervention is well established, the management of TIIEL remains controversial [12]. It is well recognized that TIIEL have subtypes with different physiopathological patterns that may explain differences in outcomes reported by studies: several procedures, ranging from direct sac puncture to complete endograft removal and conversion to open surgical repair, are reported with controversial results [9,12].

The aim of the present study is to report the outcome of patients presenting an isolated type II endoleak (TIIEL) that requires reintervention and to identify clinical and anatomical characteristics potentially implicated in refractory TIIEL occurrence and fate.

## 2. Materials and Methods

From January 2003 to December 2020, 3230 patients were submitted to standard EVAR for infrarenal AAA in two high-volume Italian tertiary referral centers: the Vascular and Endovascular Surgery Unit of the Sant’Andrea Hospital of Rome, “Sapienza” University of Rome, and the Vascular and Endovascular Surgery Unit of the San Camillo-Forlanini Hospital of Rome.

Among all patients treated within the study period, those requiring reintervention for a TIIEL were retrospectively analyzed from a prospectively collected database. Patients treated outside devices’ instruction for use (IFU) or with endovascular aneurysm sealing (EVAS) and those who were not submitted to at least two postoperative computed tomographic angiography (CTA) were excluded from the study.

Indication for AAA repair was based primarily on aneurysm diameter; rate of growth more than 1 cm/year and aortic wall morphology were also considered [13]. In both centers, indications for EVAR were based on age, comorbidities, operators’ experience, and patient preferences [14]. Prior to initial EVAR procedure, all patients underwent an extensive assessment, including clinical history reporting, physical examination, chest radiography, electrocardiography, pulmonary function testing, transthoracic echocardiography, and laboratory testing. Patients also underwent CTA of the entire thoracic and abdominal aorta to evaluate the presence of other aortic lesions and to determine the feasibility of EVAR. CTA was performed with and without contrast medium during arterial and venous phases using a 1 mm slice thickness. All measurements were performed using a workstation with dedicated software and center lumen line reconstruction (OsiriX MD software version 12; PIXMEO, Bernex, Switzerland, and Terarecon software version 4.4.8.851.2194; TeraRecon, Foster City, CA, USA). Postanalysis included 3-dimensional volume rendering, preoperative simulated angiography, and multiplanar reconstruction [15]. EVAR procedures were all performed with commercially available devices according with their IFU; endograft selection, type of anesthesia, surgical or percutaneous access, and necessity for adjunctive intraoperative procedures (sac embolization, and/or patent aortic branches embolization) were tailored for each patient.

After EVAR, the follow-up protocol included, for both centers, physical examination, duplex-ultrasound scan (DUS), and CT at 30 days. DUS was then performed at 3 and 6 months, at 1 year, and yearly thereafter. All patients underwent CT imaging 1 year after the initial procedure, without further CT examinations in the absence of complications detected by yearly DUS follow-up.

In case of a persistent TIIEL determining sac expansion greater than 10 mm, according to latest ESVS guidelines [13], patients underwent a further endovascular procedure aiming to exclude the aneurismal sac. Redo endovascular procedures were either performed via endoleak nidus direct embolization and/or aortic side branches occlusion.

TIIELs responsible for persisting aneurysmal sac perfusion 6 months after redo endovascular procedures were classified as “refractory” and submitted to open conversion due to the high risk of aneurysm expansion related complications [16,17]. Conversion was performed with or without endograft preservation according to patient status, and operators’ decision.

Complete endograft explantation was performed in a standard fashion [18]. Conversion with graft preservation (semiconversion) was performed as previously described. Via a standard retroperitoneal approach through the 11th intercostal space, without rib resection or left kidney reflection, the infrarenal aorta was prepared and a preventive banding and reshaping of the neck with a Teflon band was carried out. Once the proximal neck was secured, the sac was opened longitudinally, and the thrombus or hygromas, or both, removed. Therefore, all the identified feeding vessels were ligated, and the sac was finally sutured leaving some fenestrations to avoid repressurization with the subsequent risk of expansion and rupture [17].

The primary outcome of the present study was to evaluate concurrency rate and fate of TIIEL requiring reintervention after standard EVAR procedures. Secondary outcomes were the necessity of open conversion after TIIEL embolization, and immediate and long-term mortality, severe adverse events (SAEs) after both, complete and semiconversion. SAEs were considered as: acute myocardial infarction, respiratory failure, and acute renal dysfunction.

Clinical, anatomical, and technical features for all the procedures were noted and considered as factors potentially influencing the outcomes.

### 2.1. Statistical Analysis

The data are reported as mean and standard deviation or as absolute frequencies and percentages (%). Intergroup comparisons for each variable were performed using the Student’s *t*-test, chi squared or Fisher’s exact test; continuous variables were analyzed using a one-way analysis of variance (ANOVA), Kruskal–Wallis test. A *p* value of <0.05 was considered statistically significant. Multivariate analysis was conducted by logistic regression using the Backward method. All analyses were calculated using SPSS version 25 (IBM Corp, Armonk, NY, USA).

### 2.2. Ethical Requirements

This study complied with the principles of the Declaration of Helsinki; patients gave their consent for procedures and data collection and analysis. Due to the retrospective nature of the study, local ethical committees were only notified.

## 3. Results

Among the 3230 standard EVAR procedures performed during the study period, 1176 (34.4%) patients were treated outside devices’ specific instruction for use and per protocol excluded by the present study. Considering the remaining 2054 patients, sac shrinkage was observed in 1169 cases (56.9%), sac stability in 514 (25.02%), while a sac enlargement due to any type endoleak was recorded in 268 patients (13.07%); finally, 103 patients were lost at follow-up (5.01%).

The crude TIIEL endoleak rate in the entire study group was 25.36% (521 patients), and most of them were only present at the 1 month postoperative CTA and spontaneously resolved within the first year of follow-up. A total of 102 patients, 4.9% of the entire population, presenting a TIIEL responsible for sac enlargement were identified and retrospectively included in present study. Eighty-eight (86.27%) patients were male, the mean age was 77.32 ± 8.08 years (range 52–99), and in 72.55% of cases the ASA class was ≥3. Demographic and clinical characteristics are reported in Table 1.

Preoperative characteristics are reported in Table 2; the mean aortic diameter was 64.7 ± 14.02 mm (range 40–98), half of treated patients had a patent IMA, and 44.11% ≥3 couples of patent LAs. Five types of commercially available devices were used in the present series, and all initial EVAR procedures were performed inside the devices’ specific manufacturers’ instructions for use (Table 3). In 49 cases (48.03%) a standard EVAR procedure was completed without adjunctive maneuvers, while in the other cases IMA embolization and/or prophylactic sac embolization were performed (Table 4). After EVAR procedures, all patients were followed as per protocol; the longest follow-up reached 14 years, and the mean interval between initial EVAR procedure and TIIEL requiring reintervention detection was 54.15 ± 41.08 months (range 1–168).

All enrolled patients were initially submitted to a further endovascular procedure once TIIEL requiring reintervention was diagnosed. In detail, 57 patients underwent LAs or IMA embolization (55.87%), 42 transarterial aneurismal sac embolization (41.17%), and three (2.96%) laparoscopic ostial ligations of the inferior mesenteric artery (Table 5).

During a mean follow-up of 14.22 ± 16.57 months (7–48), the redo endovascular approach was able to ensure complete sac exclusion in 52 cases, while 50 patients presented a still evident refractory TIIEL and therefore a surgical conversion was planned.

All patients presenting a refractory TIIEL were submitted to surgical conversion, with (10 patients) or without (40 patients) endograft excision. The mean hospital stay after conversion was 10.12 ± 5.8 days, no intraoperative mortality was observed after complete surgical conversion or semiconversion, and cumulative in-hospital SAEs rate was 16% (eight patients, two after semiconversion). At a mean follow-up of 54.75 ± 39.83 (range 1–132), seven patients died (14%): two due to fatal myocardial infarction after semiconversion, and five after endograft explantation (three myocardial infarction, one lung cancer, and one aorto-enteric fistula secondary to graft infection).

Retrospectively revising our data, patients presenting a refractory TIIEL were significantly different from those who did not develop the complication in terms of preoperative clinical and morphological characteristics: coronary artery disease occurrence (*p* = 0.005, OR: 3.18, CI95%: 1.3–7.2); preoperative AAA sac diameter (*p* = 0.0055); patent IMA (*p* = 0.016, OR: 2.64, CI95%: 1.18–5.90); three or more patent LAs (Figure 1; Table 1 and Table 2).

Additionally, patients were different regarding initial EVAR procedures: those developing a refractory TIIEL were more likely to have been submitted to isolated standard EVAR without adjunctive procedures (*p* > 0.0001; OR: 9.48, CI95%: 3.84–23.4; Table 4).

Lastly, the two groups of patients were significantly different in terms of performed redo endovascular procedures, as shown in Table 5.

Multivariate analysis confirmed that preoperative AAA diameter > 65 mm, at time of the first EVAR procedure, was a predictive independent factor of refractory TIIEL occurrence, while an association of EVAR and adjunctive procedures (IMA, aneurysmal sac, and/or lumbar arteries embolization) were all independently associated with a lower risk of developing the complication (Table 6).

## 4. Discussion

According to previously published documents, TIIEL after standard EVAR performed inside IFU is not a rare event, affecting approximately 20–40% of the study population although the crude rate of TIELs requiring reintervention is much lower [9,12,19]. In a recent systematic review performed on over 33 studies and 2643 patients with TIIEL, Charisis and coworkers reported a sac expansion rate of 29% and an overall rupture rate of 1.1% [19].

Consistently, in the present study, over a period of 18 years and with more than 2000 procedures performed in two tertiary referral Italian hospitals, approximatively 25% of patients presented a TIIEL, while less than 5% of them required a reintervention due to a TIIEL-related sac expansion. Furthermore, in half of the cases, TIIEL was fixed with a new minimally invasive endovascular procedure.

Despite these apparently reassuring data, it is undeniable that the TIIELs, generally considered a relative benign disease, may evolve into a malignant disease, leading to a progressive sac diameter increase and even to an after-EVAR AAA rupture [9,19]. In this regard, several studies have tried to clarify potential risk factors (clinical, anatomical, and technical) predisposing the onset of this infrequent but disastrous occurrence. Piazza and collaborators tried to stratify the risk of TIIEL, defining patients as “at risk” when at least one of the following criteria was present: patency of a more than 3 mm IMA, or of at least three pairs of lumbar arteries, or two pairs of lumbar arteries and a sacral artery or accessory renal artery or any diameter patent IMA. In their experience, the “at risk” group had a higher likelihood of both TIIEL and TIIEL related reinterventions [20]. In a more recent experience, Ide and coworkers revealed that chronic kidney disease stage ≥ 4, patent IMA, and a number of patent LAs were risk factors of aneurysm sac enlargement caused by TIIEL, essentially confirming the previous reported results [21]. Additionally, the present series seems to confirm this hypothesis except for the role of chronic kidney disease that was not observed in our patients, while CAD was significantly associated with differences in outcome. Notably, those results were not unanimously founded in published literature [8,22].

As previously mentioned, the majority of TIIELs requiring reintervention could be effectively managed by a simple, minimally invasive endovascular procedure as elegantly reported by Chen and Stavropoulos in their recent review. However, surgical treatment still represents the standard of care for the minority of cases that have failed embolization [12].

Consequently, the leading question is how to identify patients who will develop a malignant refractory TIIEL potentially requiring a surgical conversion.

Moreover, the present experience has allowed us to highlight that some preoperative clinical or anatomical factors, pre-existing to the initial EVAR procedure, were significantly related to the development of a refractory TIIEL occurrence during follow-up. Indeed, the patients who developed a refractory TIIEL presented a higher prevalence of coronary artery disease (*p* = 0.005, OR: 3.18, CI95%: 1.3–7.2), a larger preoperative AAA sac diameter (*p* = 0.0055), and were more prone to present a patent IMA (*p* = 0.016, OR: 2.64, CI95%: 1.18–5.90) or three or more patent LAs (Figure 1; Table 1 and Table 2). Furthermore, patients with refractory TIIEL were more likely to have been initially submitted to isolated standard EVAR without adjunctive procedures (*p* > 0.0001; OR: 9.48, CI95%: 3.84–23.4; Table 4). Likewise, multivariate analysis demonstrates that a preoperative AAA diameter greater than 65 mm at time of the first EVAR procedure was a predictive independent factor of refractory TIIEL occurrence, while the association of EVAR and adjunctive procedures (IMA, aneurysmal sac, and/or lumbar arteries embolization) was independently and invariably associated with a lower risk of developing the complication (Table 6). Consistent with previous published experiences [23,24], our findings seem to endorse the hypothesis that larger AAA should be initially treated by an open approach in fit patients, or with EVAR associated with adjunctive endovascular maneuvers [25].

Furthermore, our data confirm the technical feasibility of a surgical conversion as definitive treatment for refractory TIIEL and, in accordance with previous published experiences, suggest that semiconversion should be considered less invasive and as effective as endograft removal [17,26].

Lastly, from a purely speculative viewpoint, we could hypothesize that the increased incidence of refractory TIIEL in patients with CAD, as shown by the univariate analysis and the ANOVA test results (Table 1 and Figure 1), could be related to an active thrombus remodeling after EVAR due to a more aggressive systemic atherosclerosis [27], involving plasmatic mediators [28]. Undoubtedly, this is only a research hypothesis that needs to be proved in further dedicated studies.

## 5. Limitations

Although the real predictive value of the presented results is supported by impressive statistical significance, the present study has several limitations. The first is the design of the study: a retrospective single-arm registry, conducted on a nonrandomized and relatively small cohort, not allowed to compare the results with a control patient population. Moreover, the retrospective nature of this study did not allow a subgroup analysis on important variables such as pre- and postoperative pharmacological anticoagulant therapy. Lastly, the mean follow-up duration, although certainly not negligible, was extremely uneven among enrolled patients.

## 6. Conclusions

According to present results, it is reasonable to try to preoperatively identify those patients who will develop a refractory TIIEL after EVAR. Above all, present data suggest the possibility of distinguishing patients requiring a minimally invasive endovascular redo from those requiring a surgical conversion.

## Figures and Tables

**Figure 1 jpm-12-00339-f001:**
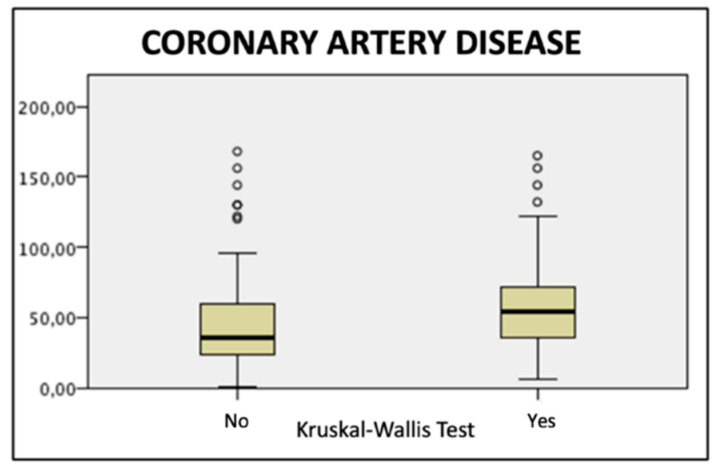
One-way analysis of variance (ANOVA), Kruskal–Wallis test, showing a highly statistical association (*p* = 0.00023) between coronary artery disease and refractory TIIEL occurrence.

**Table 1 jpm-12-00339-t001:** Preoperative demographic and clinical characteristics of patients included in present study.

	TIIEL Requiring Reintervention (102 Patients)	TIIEL Solved after Redo Endovascular Procedure (52 Patients)	Refractory TIIEL (50 Pts)	*p* (OR; CI95%)
Mean Age (years ± SD)	77.32 ± 8.08	77.42 ± 7.62	77.22 ± 8.60	0.449909
Male Sex (*n*; %)	88; 86.27	43; 82.69	45; 90	0.28 (0.53; 0.16–1.71)
Hypertension (*n*; %)	82; 80.39	38; 73.08	44; 88	0.057 (0.37; 0.12–1.05)
CAD (*n*; %)	41; 40.19	14; 26.92	27; 54	0.005 (3.18; 1.3–7.2)
Dyslipidemia (*n*; %)	39; 38.23	20; 38.46	19; 38	0.90 (1.01; 0.45–2.26)
Smoking History (*n*; %)	32; 31.37	18; 34.62	14; 28	0.47 (1.36; 0.58–3.15)
COPD (*n*; %)	27; 26.47	16; 30.77	11; 22	0.31 (1.57; 0.64–3.84)
Diabetes (*n*; %)	19; 18.62	10; 19.23	9; 18	0.87 (1.08; 0.39–2.92)
ASA Classification				
I (*n*; %)	0	0	0	-
II (*n*; %)	28; 27.45	15; 28.85	13; 26	0.06 (0.40; 0.15–1.07)
III (*n*; %)	56; 54.9	25; 48.08	31; 62	0.15 (0.56; 0.25–1.24)
IV (*n*; %)	18; 17.65	12; 23.08	6; 12	0.14 (2.2; 0.75–6.41)

**Table 2 jpm-12-00339-t002:** Preoperative anatomical characteristics of patients included in present study.

	TIIEL Requiring Reintervention (102 Patients)	TIIEL Solved after Redo Endovascular Procedure (52 Patients)	Refractory TIIEL (50 Pts)	*p* (OR; CI95%)
Preoperative AAA diameter (mm ± SD)	64.7 ± 14.02	65.43 ± 11.96	73.3 ± 16.71	0.0055
Patent IMA (*n*; %)	55; 53.92	22; 42.31	33; 66	0.016 (2.64; 1.18–5.90)
Patent lumbar arteries				
1 couple (*n*; %)	7; 6.87	4; 7.69	3; 6	0.73 (0.76; 0.16–3.6)
2 couples (*n*; %)	50; 49.01	40; 76.92	10; 20	>0.0001 (0.075; 0.02–0.19)
3 couples (*n*; %)	37; 36.27	7; 13.46	30; 60	>0.0001 (9.64; 3.63–25.6)
>3 couples (*n*; %)	8; 7.84	1; 1.92	7; 14	0.02 (8.30; 0.98–70.1)

**Table 3 jpm-12-00339-t003:** Details of implanted endografts in present series.

	TIIEL Requiring Reintervention (102 Patients)	TIIEL Solved after Redo Endovascular Procedure (52 Patients)	Refractory TIIEL (50 Pts)	*p* (OR; CI95%)
Excluder (*n*; %)	52; 50.98	19; 36.53	33; 66	0.017772 (11.9441)
Endurant (*n*; %)	14; 13.72	9; 17.31	5; 10	
Zenith (*n*; %)	24; 23.52	14; 26.92	10; 20	
Ovation (*n*; %)	10; 9.8	9; 17.31	1; 2	
Talent (*n*; %)	2; 1.98	1; 1.93	1; 2	

**Table 4 jpm-12-00339-t004:** Adjunctive procedures performed during initial EVAR in present series.

	TIIEL Requiring Reintervention (102 Patients)	TIIEL Solved after Redo Endovascular Procedure (52 Patients)	Refractory TIIEL (50 Pts)	*p* (OR; CI95%)
Isolated standard EVAR	49; 48.03	12; 23.08	37; 74	>0.0001 (9.48; 3.84–23.4)
EVAR + sac embolization	28; 27.45	18; 34.62	10; 20	0.09 (0.47; 0.19–1.15)
EVAR + IMA embolization	25; 24.52	22; 42.31	3; 6	>0.0001 (0.08; 0.02–0.31)

**Table 5 jpm-12-00339-t005:** Redo endovascular procedures performed to treat persistent TIIEL in present series.

	TIIEL Requiring Reintervention (102 Patients)	TIIEL Solved after Redo Endovascular Procedure (52 Patients)	Refractory TIIEL (50 Pts)	*p* (OR; CI95%)
IMA embolization	23; 22.54	14; 26.92	9; 18	0.28 (0.59; 0.23–1.53)
Sac embolization	42; 41.17	7; 13.46	35; 70	>0.0001 (15; 5.51–40.7)
Lumbar arteries embolization	34; 33.33	28; 53.84	6; 12	>0.0001 (0.11; 0.04–0.32)
IMA laparoscopic ligation	3; 2.96	3; 5.78	/	0.08 (NA)

**Table 6 jpm-12-00339-t006:** Multivariate analysis (Backward method) on factor related to refractory TIIEL occurrence in present series.

Refractory TIIEL			
Included Variables			Excluded Variables
	*p*	(OR; CI95%)	
Preoperative AAA diameter > 65 mm at first EVAR procedure	0.03	(11.7; 1.27–107.9)	Patent IMA
IMA embolization	0.0002	(0.0043; 0.0002–0.077)	CAD
Sac embolization	<0.0001	(0.0013; 0.0001–0.0210)	2 couples
Lumbar arteries embolization	0.0009	(0.009; 0.0006–0.14)	3 couples
			>3 couples
			Isolated standard EVAR
ROC curve analysis			
Area under the ROC curve (AUC)		Standard Error	CI95%
0.96		0.017	0.91–0.99

## Data Availability

The data presented in this study are available on request from the corresponding author. The data are not publicly available due to their containing information that could compromise the privacy of research participants.

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
