# Peer review of "Incidence and Fate of Refractory Type II Endoleak after EVAR: A Retrospective Experience of Two High-Volume Italian Centers"

_jpm, 2022, doi:10.3390/jpm12030339_

Round 1

Reviewer 1 Report

Sirignano P., et al, in their study carry out a retrospective experience of Incidence and Fate of Refractory Type II Endo-2 Leak After Evar of two high volume Italian centers.

The study is well conducted and looking at multiple centers add to the strength of the study. The writing could be modified for an easy understanding at transition.

Author Response

The Authors would thank the Reviewer for his/her kind words and valuable comments.

Where applicable, text was modified according to his/her suggestions.

Reviewer 2 Report

The article presents the outcomes of patients with a type 2 endoleak after an EVAR procedure, aiming to identify the characteristics implicated in refractory TIIEL.

Although the study is retrospective, it provides valuable data on the real life outcomes and management of 102 patient with TIIEL requiring reintervention. The results are supported by robust statistical diferences.

I have some concerns regarding the heterogeneity within the follow up period after the re-do intervention, with a wide range.

I recommend this article for publication

Author Response

The Authors would thank the Reviewer for his/her kind words and valuable comments.

We agree with Reviewer’s concern regarding the heterogeneity of the follow-up period after the re-do intervention. Unfortunately, the retrospective nature of the study does not allow to have a uniform follow-up for all treated patients. According with Reviewer’s suggestion a sentence was added in study’s limitations to clearly discuss this issue.
